# The "Coherent Data Set": Combining Patient Data and Imaging in a Comprehensive, Synthetic Health Record

Jason Walonoski [ID], Dylan Hall, Karen M. Bates *[ID], M. Heath Farris, Joseph Dagher, Matthew E. Downs, Ryan T. Sivek, Ben Wellner, Andrew Gregorowicz, Marc Hadley, Francis X. Campion, Lauren Levine, Kevin Wacome, Geoff Emmer, Aaron Kemmer, Maha Malik, Jonah Hughes [ID], Eldesia Granger [ID] and Sybil Russell

The MITRE Corporation, Bedford, MA 01730, USA; jwalonoski@mitre.org (J.W.); dehall@mitre.org (D.H.); hfarris@mitre.org (M.H.F.); jzdagher@gmail.com (J.D.); medowns@mitre.org (M.E.D.); rysivek@gmail.com (R.T.S.); wellner@mitre.org (B.W.); andrewg@mitre.org (A.G.); mhadley@mitre.org (M.H.); fcampion@mitre.org (F.X.C.); lelevine@mitre.org (L.L.); kwacome@mitre.org (K.W.); gemmer@mitre.org (G.E.); aakemmer@mitre.org (A.K.); mmalik@mitre.org (M.M.); jphughe2@ncsu.edu (J.H.); egranger@mitre.org (E.G.); srussell@mitre.org (S.R.)
* Correspondence: kbates@mitre.org

**Abstract:** The "Coherent Data Set" is a novel synthetic data set that leverages structured data from Synthea™ to create a longitudinal, "coherent" patient-level electronic health record (EHR). Comprised of synthetic patients, the Coherent Data Set is publicly available, reproducible using Synthea™, and free of the privacy risks that arise from using real patient data. The Coherent Data Set provides complex and representative health records that can be leveraged by health IT professionals without the risks associated with de-identified patient data. It includes familial genomes that were created through a simulation of the genetic reproduction process; magnetic resonance imaging (MRI) DICOM files created with a voxel-based computational model; clinical notes in the style of traditional subjective, objective, assessment, and plan notes; and physiological data that leverage existing System Biology Markup Language (SBML) models to capture non-linear changes in patient health metrics. HL7 Fast Healthcare Interoperability Resources (FHIR®) links the data together. The models can generate clinically logical health data, but ensuring clinical validity remains a challenge without comparable data to substantiate results. We believe this data set is the first of its kind and a novel contribution to practical health interoperability efforts.

**Keywords:** synthetic data; biological models; electronic health records; personally identifiable information

## 1. Introduction

The healthcare industry operates under strong data privacy and data security restrictions [1,2]. The inaccessibility of health data has created a state of health data poverty that impacts innovation, including software development and integration, clinical and public health research, and the application of machine learning [3]. Due to a lack of generally available and publicly sharable data, the healthcare industry suffers from a corresponding lack of interoperability, including data exchange and data semantic issues [4].

These issues are compounded by the fact that healthcare data exist in a wide variety of forms (e.g., structured relational data, unstructured free-text clinical notes, time-series such as electrocardiograms (ECG), images such as magnetic resonance imaging (MRI), and biomedical data such as genome sequences) from a wide variety of sources (e.g., electronic health care records, laboratories, pharmacies, devices, wearable sensors, and insurance claims) that are not typically integrated [5].

De-identification (also called anonymization) has been used to make health data available, but research has shown that de-identification does not guarantee privacy or eliminate risk [6,7]. Synthetic data have been used as an alternative to de-identification. Synthetic data are generated either by building models from aggregated statistics (given the assumption that real data are protected and not readily available) [8] or more frequently

by training models with real individual data instances (given the assumption that real data are accessible, which ignores that accessibility is part of the problem to begin with) [9–11]. Additional prior research has focused on the generation of medical images using adversarial training [12,13], which accounts for a vast majority of synthetic data research, as well as on generating sets of attributes [14], clinical notes [15], and genomics [13,16].

There have been prior attempts to create synthetic health data that focus on interoperability [8,17] as well as work that attempted to enrich such data with de-identified time-series data (e.g., ECGs recorded in an ICU) [18] and work that linked together synthetic brain imagery phenotypes and synthetic genomic data to address the complexity of real health data that exist in various forms [13].

We believe this paper is the first that combines a multiplicity of synthetic data forms together in a single package and makes it publicly available as a digital public good. With a focus on Cardiovascular Disease (CVD), the Coherent Data Set combines synthetic genomics, imaging, clinical notes, and physiology data together into a complete profile. CVD was selected because it is a commonly occurring disease with a high prevalence rate and severe health outcomes that represents a significant global burden [19].

The value of the Coherent Data Set and its contribution to the field of synthetic data in healthcare is in its novel combination of distinct types of healthcare data. Learnings from the Coherent Data Set are not necessarily to be derived from a quantitative or qualitative analysis, but from a study of how the data elements join to form a comprehensive profile of entirely synthetic patients. This data set was produced by and for researchers who require broadly scoped and integrated data sets but are unable to access such information without putting patient privacy at risk.

The purpose of this research is to make publicly available to industry and researchers realistic-but-not-real synthetic health data that conform both to international standards (such as HL7 Fast Health Interoperability Resources [FHIR]) and to clinical terminologies [20]. The overarching goal of this research is to further health data interoperability. The purpose of this paper is to document the process and methods used to generate the Coherent Data Set, to document the data it includes, and to make this resource widely available to the community. In the Materials and Methods section, we describe the relevant techniques used to generate each of the datatypes in the Coherent Data Set. In the Results section, we summarize the final synthetic data set that has been constructed and is available for download (with additional materials in the Appendix A and Supplementary Materials). We explain the significance of the work, the suitability of the data for various use-cases, and the overall pros and cons of the data set in the Discussion section. We conclude the paper with a summary and our final thoughts on the Coherent Data Set.

## 2. Materials and Methods

### 2.1. The Coherent Data Set

To emulate realistic patient populations, the Coherent Data Set is comprised of patients with Cardiovascular Disease (CVD) and those who exhibit risk factors for the disease. According to the CDC, roughly 655,000 Americans die from CVD every year; this is the equivalent of 1 in 4 deaths in the United States [19]. The common prevalence and physiological complexity of CVD make the class of diseases ideal for modeling within the Coherent Data Set, since the objective of the resource is to support developments in interoperability. Data with realistic structures, relationships, and accurate representations of health concepts are necessary components for improving healthcare data interoperability.

The Coherent Data Set leverages structured data produced by the open-source patient population simulator Synthea. Synthea uses models of disease progression and treatment to simulate the clinical history of synthetic patients and export the resulting synthetic data in standardized data formats, most notably FHIR. The synthetic data are considered to be free of privacy and security risks [8].

The Coherent Data Set contains several types of information typically captured by clinical providers caring for CVD patients: genomic data, MRI imaging, clinical notes, and

physiological data. Synthea provides the foundational structured electronic health records (EHRs) that join each of these data elements for a comprehensive synthetic patient profile. Each time a synthetic patient experiences an event that generates an electrocardiogram (ECG), an MRI brain scan, or genomic lab report data, the new data are merged with the existing information in their EHR using the following HL7 FHIR resources: Observation, Procedure, ImagingStudy, DiagnosticReport, DocumentReference, and Binary.

The resulting data emulate longitudinal and mixed-modality health data contained in EHRs. Realistic synthetic data can be used for software development, data analysis, capacity planning, and to understand the potential impacts of different policies on health outcomes. The Coherent Data Set can also be used to model health care records, clinical data, and diagnostic reports for research purposes where realistic-but-not-real data are sufficient, without the risks to privacy that arise with the use of health data from real patients.

Figure 1 shows how the various data types and models interact to trigger additional data generation and information connections as external additions to Synthea.

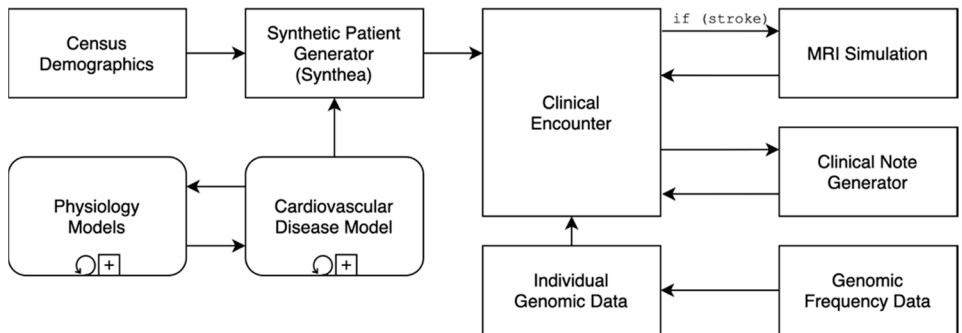

**Figure 1.** Concept map for the Coherent Data Set. The synthetic patient generator takes census demographics as input to generate a set of patients. The generator includes a cardiovascular disease model simulating the progression and treatment of cardiovascular disease, which in turn includes a physiology model to simulate the cardiovascular system and produce artifacts such as ECGs. The generator produces patients with a series of clinical encounters. Each clinical encounter may optionally include MRI images (e.g., if the patient had a stroke), clinical notes, and genomic data (e.g., if a genetic lab panel was ordered).

The following subsections describe the methods and techniques used to generate each of the component data types: genomics, imaging, clinical notes, and physiology data. The overall results are presented in the Results section.

### 2.2. Genomics

Several genetic markers are clinically linked to CVD. To better model the impact of genetic markers attributed to the disease, a database of genetic risk alleles and their CVD linkages was developed from published clinical studies (Supplementary Table S1). Using publicly available human sequence profiles from the 1000 Genomes Project database, a second database of genomic profiles for CVD risk alleles was created from human genome data [21]. This second database was used to calculate the rate of co-occurrence for the risk alleles within a given haplotype for each human sequence profile, generating an estimation of allele frequency when a synthetic genome is created. Global allele frequencies for the CVD risk alleles were used to estimate population frequencies.

To create the genomic profiles for the CVD risk alleles database, two genome scaffolds containing CVD risk allele profiles were used to generate the necessary synthetic genome sequence data. Subsequently, gamete pools were created for individuals in the patient population, and the model determined the recombination and co-occurrence frequencies for the disease alleles. Gametes were combined from each parent using a random mating selection, resulting in the filial 1 or F1 generation. This process continued with new alleles injected into the lineage to produce the F2 and then F3 generations.

Prior to this work, Synthea only contained a few genetic markers in the cancer modules that were probabilistically assigned to the synthetic patients. With the CVD risk allele analysis and a human reference genome from the 1000 Genomes Project database, a synthetic genome was created with a genetic family history for three generations. Appendix A Figure A5 illustrates this process. Every individual in the synthetic genome has a genome sequence file in FASTA format: a representation of their sequence in single-letter nucleotide codes. The synthetic genome is created using the human reference genome (version GRCh37.p12) as a scaffold. Within the genome, variant alleles of the parental generation replace the reference alleles across the genome to create the complete synthetic genomic profile. Only the selected variants for each parent are modified. To produce familial synthetic genomes, the parental genomes are broken by chromosome. A copy of each chromosome is passed from each parent to the progeny, simulating genetic reassortment. The synthetic genome of the progeny is combined into a FASTA file.

The Population Algorithm builds on the synthetic genome to model the distribution of genetic markers, particularly those that indicate risk for CVD (Supplemental Table S1), in a selected population. First, synthetic data are generated by the Synthea probabilistic models, and then synthetic patients with a record of CVD-related conditions are randomly selected to have genetic testing profiles generated for them. The selected subpopulation determines co-occurrence and, in combination with the size of the given subpopulation, the attributes define the anchoring mutations. A complete, 100 percent co-occurrence frequency for single nucleotide polymorphisms (SNPs) is included. The SNPs are ordered by subpopulation of co-occurring frequencies for two alleles, and then the ordering is repeated for one allele [21]. This process creates an individual CVD SNP profile. Validation of the synthetic genomes and the distribution of genetic markers within the resulting population ensured the integrity of the genomic sequence and the SNP occurrence across the population, representative of frequencies observed within the 1000 Genomes Project database [22]. Appendix A Figure A6 illustrates this process.

The Coherent Data Set and genomic information also enable synthetic diagnostic lab reports, which are a reproduction of a diagnostic panel alongside genomic testing results that can be added to the EHR of a patient. Figure 2 illustrates an instance in which a synthetic patient received genetic testing, and their genetic profile was added to their health record. The synthetic genome data and the Population Algorithm are important foundations that can serve as building blocks to expand upon the existing data in the Coherent Data Set, should future work involve adding new diseases or data types.

The approach described for synthetic patients with CVD could be used as a model for expansion to a clinical genomic disease markers database. While the current focus is on CVD, the CVD risk allele analysis and database could be expanded to include other diseases, supporting research in additional areas. Following the demonstration of the concept for CVD, the first steps would require the documentation of the genetic risk markers associated with the given disease cluster, the creation of a synthetic disease population, and the tracking of allele propagation through mating and recombination events. While the current study leveraged the SNPs of the 1000 Genomes Project database with 2504 individual genomes from around the globe, more sizeable datasets containing even larger stores of SNP data could be used to enhance the methodology and provide greater fidelity of the SNP data representing the zygosity of the global population.

The Population Algorithm could also be updated to include filtering or selection options for randomly sized populations beyond the family, such as a patient group or community. Appendix A Figure A6 illustrates how this expansion in population grouping could show alleles throughout the defined community; this may be useful for modeling population health variables. Another potential area for expansion is the synthetic diagnostic lab reports. Increasing the size and scope of the genetic reports could allow for queries to generate or request synthetic genomic data for families or populations.

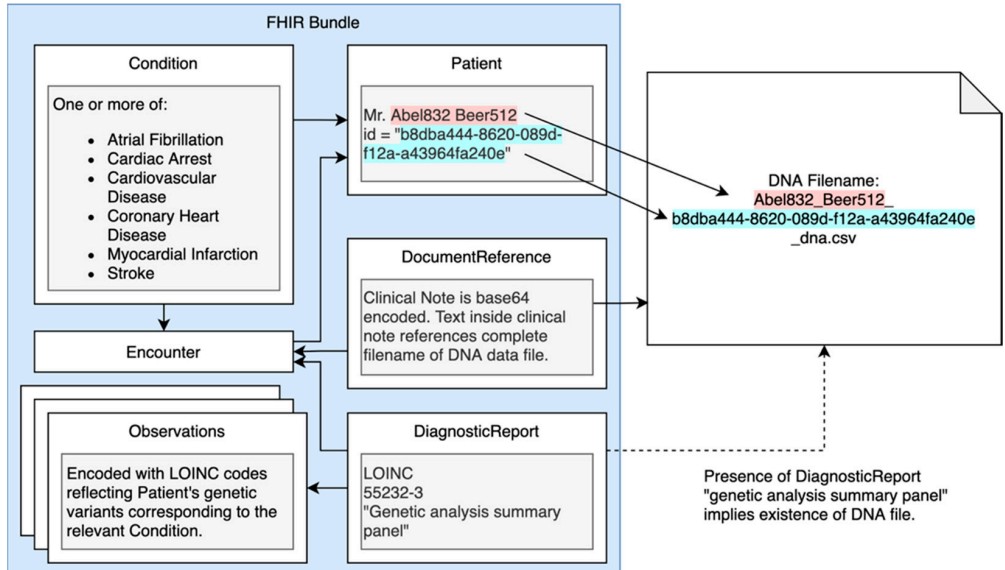

**Figure 2.** The relationship of FHIR Resources to the synthetic DNA data. A diagnostic report and observations relating to a synthetic genome of the patient are generated during an encounter and then added to their health record. The DNA data file is referenced by a DocumentReference resource within the HL7 FHIR Patient Bundle.

### 2.3. Imaging

CVD can have a significant impact on a patient's neurological health as well. The most common cardiac-related neurological injury is a cardiac embolism, but recent studies have also linked both Alzheimer's and Parkinson's diseases to CVD [23]. Synthetic Magnetic Resonance Imaging (MRI) brain scans, which are frequently taken after a stroke, were created using a numerical "phantom" simulation to model the neurological effects of CVD-related health events. Prior to the development of the Coherent Data Set, Synthea only contained references to imaging in the structured EHR data, rather than any visual components. With the incorporation of synthetic MRI images in the Coherent Data Set, three areas of focus emerged: developing a numerical model of the human head with and without the presence of pathology, developing a mathematical model for a rapid magnetic resonance (MR) image simulator, and developing the capability to generate unique brain images representing different subjects.

Numerical and physical "phantoms" are the standard techniques for validating novel MRI protocols [24]. While physical phantoms can be utilized to calibrate MRI scanners and protocols, they require the use of and access to an MRI scanner. Numerical phantoms allow for the development and testing of those novel protocols without the complications associated with obtaining access to an MRI scanner or the costs associated with the scans. Synthetic phantoms are easily modifiable according to user specifications, instead of a single composition as with the physical phantom.

Prior MRI numerical phantoms of the brain were derived from a single patient, and the variation of the tissue properties and structures was limited [25,26]. This spurred the development of our numerical phantom to capture unique brain structures and tissue properties for each individual subject. The development of the numerical phantom was completed in three stages: a Root Segmentation Map (RSM), followed by a Subject Segmentation Map (SSM), and lastly, electromagnetic tissue properties (EMTP) were assigned to the voxel representation. EMTPs are the basis of the signal contrast seen in the MR image space and are chosen based on typical and pathological conditions.

The process of developing the RSM is illustrated in Appendix A Figure A7. A T1-weighted MRI scan of a human head (T1-weighted, 465 image size isotropic, 0.4 mm isotropic resolution) was segmented into 18 brain regions (frontal cortex, motor cortex,

parietal cortex, occipital cortex, temporal lobe, white matter, cerebellum, cerebellum white matter, caudate, putamen, thalamus, amygdala, hippocampus, red nucleus, substantia nigra, globus pallidus, temporal claustrum, and brainstem) as well as cerebral spinal fluid (CSF), skull, soft tissue, eyes, and sinus cavities/air. The FreeSurfer Image Analysis Suite and MATLAB were leveraged for automatic and manual segmentation, respectively [27].

The RSM was used to generate Subject Segmentation Maps (SSMs) by adjusting the boundaries of RSM brain structures and adding pathology if required. The boundary adjustments can be applied to all head structures or to structures specified by the user.

The final step was populating the regions with a set of corresponding EMTP tissue properties. These EMTP tissue properties were derived from T1, T2, and T2* MRIs, and the mean and standard deviation of iron concentration values. EMTP tissue properties were collected through a comprehensive peer-reviewed literature survey conducted on PubMed, Web of Science, and Google Scholar [28–34]. Values for both healthy subjects and patients with neurological disease were collected, ranging in age from 0 to 100 years old.

The output EMTP map was then processed through an MRI simulator to generate DICOM imaging files to attach to the associated subject in the Coherent Data Set. The DICOM files included both the magnitude and the phase output from the MRI simulator, as some cardiac pathologies are only detected on phase imaging compared to the traditional magnitude image.

The MRI simulator and output have been conditionally validated against prior studies and physical susceptibility phantoms [35,36]. The tissue properties and structures were derived from prior histological and structural studies [37,38]. Full validation against a human scan is challenging due to individual variations in subject brains and would require exact alignment. Thus, the validation of the model is limited until a physical phantom is manufactured with specifications defined by the numerical phantom.

The MRI imaging simulator, the third area of focus for imaging development in the Coherent Data Set, can augment synthetic patient data with corresponding MRIs with associated pathology noted in the Coherent Data Set, as illustrated in Figure 3.

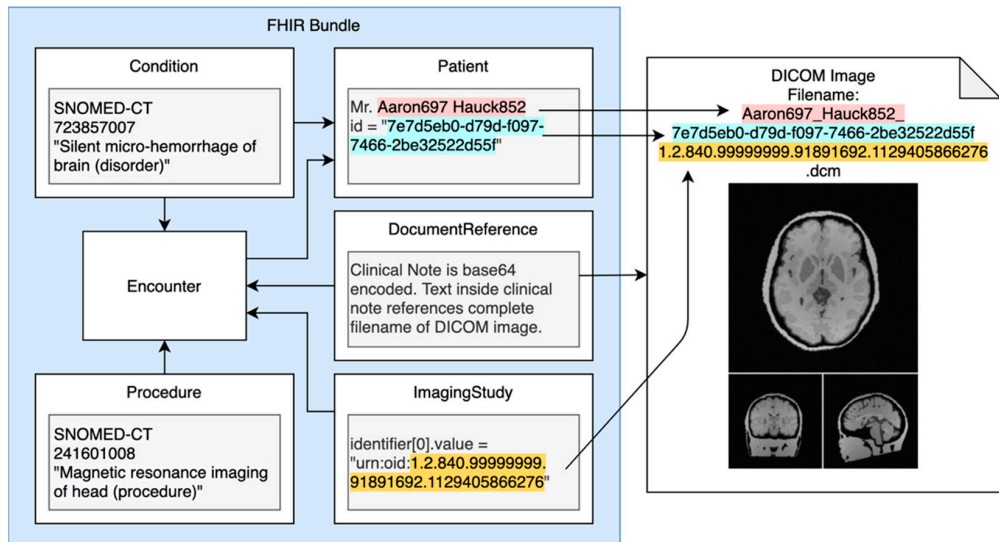

**Figure 3.** Relationship of FHIR Resources and a DICOM image file. An encounter for a synthetic patient that received imaging includes the procedure and imaging study and a reference to the external DICOM image file.

### 2.4. Clinical Notes

Every patient encounter, genetic consultation, or imaging scan generates knowledge in the form of clinical notes. Four types of synthetic clinical notes were drafted and considered for inclusion in the Coherent Data Set: clinical progress, echocardiogram (ECHO), ECG, and radiology notes. The notes were initially generated from a foundation of initial clinical notes

that used a Long Short-term Memory (LSTM) model trained with MIMIC data [39]. Additional models, including GPT-2, Transformer-XL, and other transformer-based language models, were trained and primed with assessment- and plan-style notes [40].

Medical professionals reviewed these synthetic clinical notes and indicated significant variations in the authenticity of their format and content. Since the notes did not align well with the rest of the synthetic data of each patient, an alternative approach was employed with template-driven notes where relevant clinical data "filled in the blanks." The templated notes are still stylized as subjective, objective, assessment, and plan (SOAP) style notes and are linked to the corresponding EHR of each patient using HL7 FHIR Document Reference and Binary FHIR resources. Figure 4 illustrates the flow of information generated during a patient encounter and the process of capturing this information in the clinical notes for a synthetic patient.

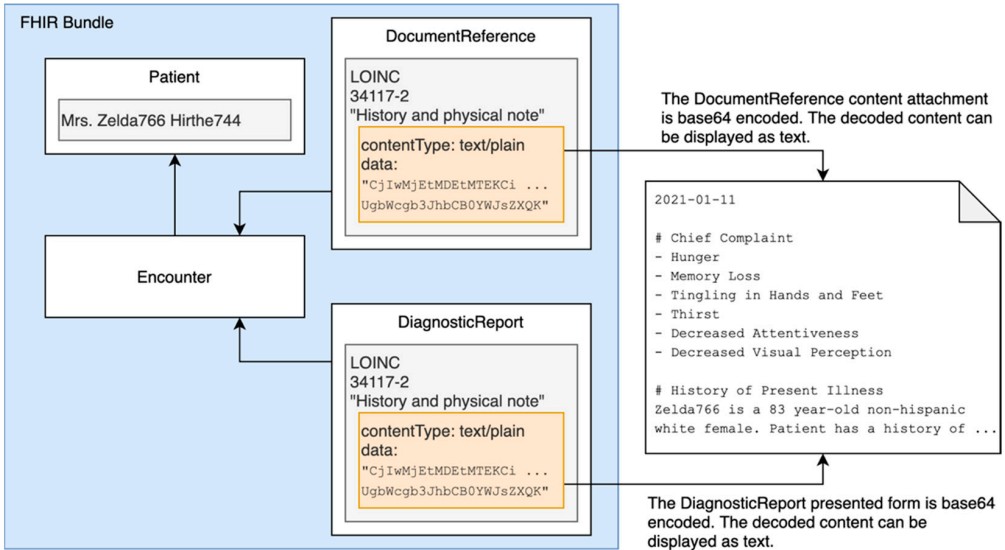

**Figure 4.** Relationship of FHIR Resources to clinical notes generated during an encounter for a synthetic patient. Rather than external files containing the notes, the notes themselves are Base64 encoded and included within the relevant FHIR resources, as per the FHIR specification.

More complex clinical notes generated with GPT-2 and trained with MIMIC data (or other real-world data sets) could offer an opportunity for non-clinical secondary use research, such as training natural language processing models. Future work could also include adding models with more structured information as part of the conditioning context to improve these techniques. Appendix A Figures A8 and A9 contain examples of templated clinical notes for synthetic patients.

### 2.5. Physiology

The Coherent Data Set expanded upon the capabilities of Synthea to include the modeling of physiological processes at the cell, organ, and organ system levels. Previously, such information was pulled from lookup tables, predefined ranges of values based on normal clinical ranges, or from values associated with stages of disease. Leveraging patient status as input and output variables in a set of equations, rather than relying upon a preset range of values, increases the complexity and allows for more sophisticated physiology modeling. An existing single-node ECG model from McSharry, et al., was used to simulate ECGs, and a model from Smith, et al., was leveraged for cardiovascular blood pressure and volumes [41,42]. The Coherent Data Set can generate realistic health outcomes data with non-linear effects as well as time-series data for tests such as an ECG, an EMG, and hemodynamics analysis. Figure 5 illustrates the process of generating and then adding an ECG for a patient to their synthetic health record.

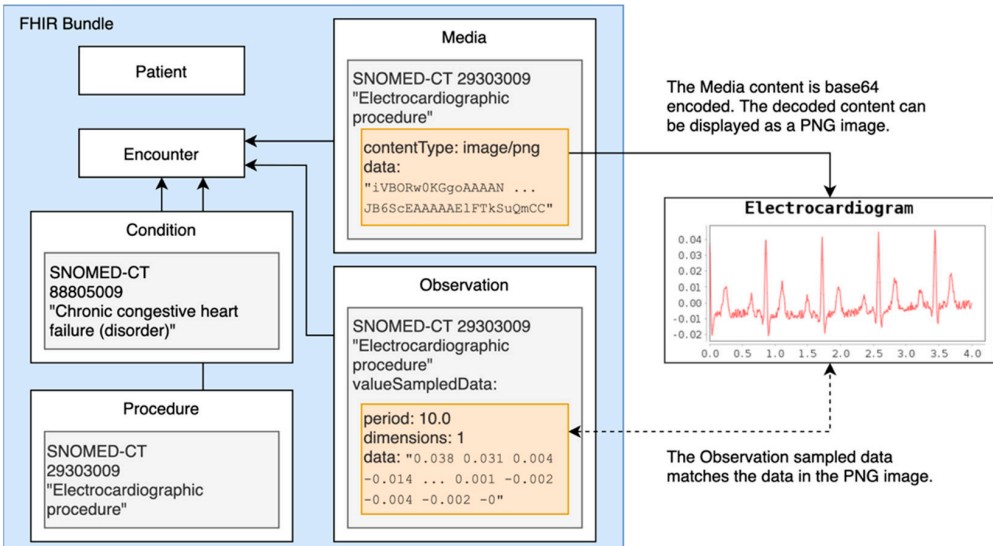

**Figure 5.** An electrocardiogram and associated observations are added to the synthetic health record of a patient. The relationships between the FHIR Resources and the ECG image are shown, where the image is a Base64 encoded PNG image included within the relevant FHIR Resources, as per the FHIR specification.

The technical development and implementation of physiological data required the addition of a new type of state in the Generic Module Framework (GMF) of Synthea. The GMF provides the basis and building blocks for disease modules to be defined as finite-state machines, with states and transitions that process and trigger patient conditions, encounters, medications, and other clinical events [8]. New constructs that instantiate and process the physiological simulations were also created to add physiological data. Systems Biology Markup Language (SBML) was selected to define the model behavior. Using a generic language such as SBML to define behavior, rather than implementing the simulation classes individually in Java, created the opportunity to incorporate and reuse existing models that can operate within the structured format. Defining the model behavior in SBML enabled the nearly immediate use of hundreds of freely licensed, existing physiological model definitions [43]. In Synthea, a new PhysiologySimulator class acts as a wrapper for the modules and contains all the logic for loading SBML files, instantiating models, supplying models with inputs based on Synthea patient attributes, running models, and providing the necessary outputs.

Physiological simulations that are particularly helpful for modeling outcomes related to CVD include hypertension, heart failure, and enhanced cardiac hemodynamics with arrhythmia effects. By adjusting the parameters of the model, Synthea can also generate several types of arrhythmias to model an ECG. Each of these events can be carried out for an individual patient to model cases based on the physiological status of the patient.

Throughout the development of the physiology models, openly available peer-reviewed models in common exchange formats from McSharry, et al., and Smith, et al., were identified, translated into SBML, and incorporated into the Coherent Data Set [41,42]. Models were selectively chosen and added to the core repository to ensure that runtime efficiency was not prohibitively impacted. The physiology simulations slow the processing rate of Synthea by approximately 50 percent. If more models were added, then running the simulations for 1000s or 10,000s of synthetic patients could push the runtime of Synthea into days or weeks rather than hours.

This effort focused on translating and integrating existing physiology models rather than developing new ones, thus requiring the establishment of face validity to ensure the results from running the models within Synthea matched the published descriptions of those models. The physiological effects were further validated and verified against clinical data using Table 1 from Hosseini, et al. [44]. Results were also shared with a subject matter

expert to ensure that they were in a realistic range and reflective of clinical statistics. The subject matter expert also reviewed the graphed values and verified that the shape of the graphed curves and the spacing of the data points emulated real clinical data visualizations for patients with cardiovascular conditions.

**Table 1.** Count of files by data types for all synthetic patients in the Coherent Data Set.

| Data Type | Count |
|---|---|
| DICOM Image | 298 |
| DNA Data | 949 |
| FHIR Bundle | 1346 |

Additionally, more sophisticated links between the genomic data and physiological simulation parameters were added to the Coherent Data Set. Dozens of genetic risk factors have been identified for various diseases, including CVD, but the precise physiological result of these genes is not always clear. Determining the most appropriate levels of physiological response to genetic markers for use in a model is a larger research question, and future developments in the field will impact the performance of the model.

To model the physiology of CVD among synthetic patients in the Coherent Data Set, simulations of commonly co-occurring conditions such as hypertension, heart failure, and arrhythmias were generated. An ECG can also be simulated using the published Synthea ECG model, based on McSharry, et al. [41]. This data can be exported as an image or as time-series data, as seen in the sample ECGs in Figure 6.

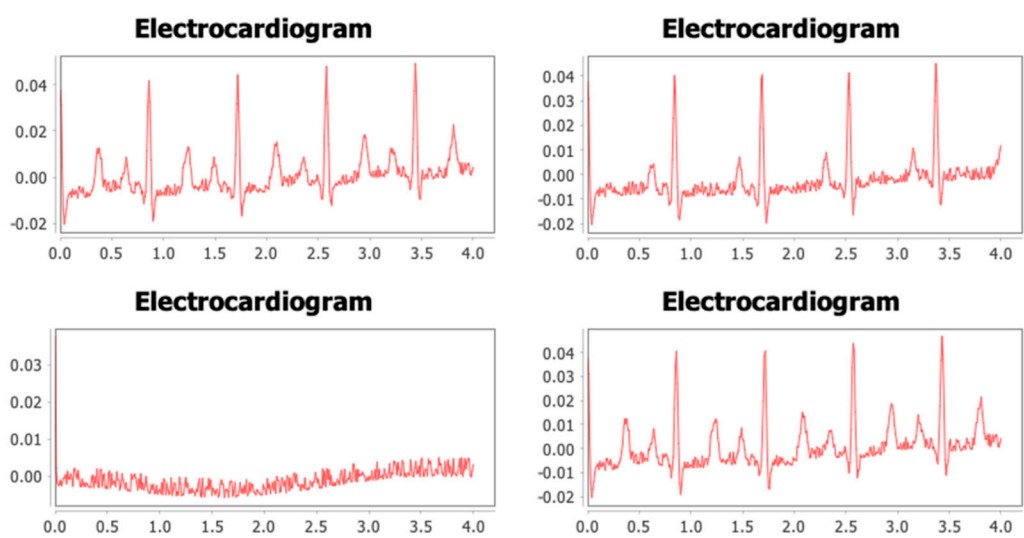

**Figure 6.** Four examples of synthetic single-node ECGs incorporated into the synthetic patient records. Examples show a clinically normal reading (**upper left** and **lower right**), non-specific ST-T wave changes (**upper right**), and the absence of a sinus rhythm reading (**lower left**). The readings contain significant artifact, which can also appear in non-synthetic ECGs due to patient movement, electromagnetic interferences, or electrode malfunctions.

Future work may include adding treatment impact to the physiology framework. The inclusion of treatments and subsequent clinical events could illustrate how a therapeutic intervention changes the physiological status of a patient. The model could enable users to see the impact of medication or other interventions on the condition of a patient; for example, what physiological changes would occur with various doses of a cardiac medication or a dietary change? Either mathematical, pharmacokinetic, or pharmacodynamic modeling could be used to indicate physiological impacts. Modeling these impacts will require significant validation work, particularly for patients with multiple comorbid conditions.

## 3. Results

The Coherent Data Set is a completely synthetic data set that links together genetic markers, MRI images, rudimentary clinical notes, and physiology waveforms into a single set of clinically logical, individual-level patient records. Comprised of more than 1000 synthetic patients, the data set contains even more patient-related imaging and data files and leverages hundreds of thousands of FHIR Resources. Table 1 shows that nearly 300 DICOM images were generated, as well as 1346 FHIR Bundles. Most synthetic patients that have a DICOM image associated with their patient profile also have a DNA data file, and all of them have an FHIR Bundle, as shown in Table 2. The total count of patients with a all three different data file types, DICOM Image, DNA Data, and FHIR Bundle, compared to patients without a DICOM image of DNA file to accompany the FHIR bundle is shown in Appendix A, Table A1.

**Table 2.** Count of synthetic patients with two or more data files.

| Data Type | DICOM Image | DNA Data | FHIR Bundle |
|---|---|---|---|
| DICOM Image | 298 | 172 | 298 |
| DNA Data | 172 | 949 | 931 |
| FHIR Bundle | 298 | 931 | 1346 |

Table 3 contains the total count of FHIR resources that are utilized to generate the synthetic patients in the Coherent Data Set. Each resource is a building block that contains patient data in a uniform format so that it can be combined with other resources and exchanged between health information systems. Since the Coherent Data Set is comprised of patients with CVD or who are at risk of developing CVD, a condition that typically requires care from many different providers, the data set contains many different resources across the health care spectrum.

**Table 3.** Count of FHIR Resources utilized in the Coherent Data Set.

| FHIR Resources | Count |
|---|---|
| Observation (includes genetic variants and ECG sampled data) | 712,512 |
| Claim | 379,511 |
| MedicationRequest | 225,996 |
| DiagnosticReport (includes Clinical notes and Genetic Panels) | 213,729 |
| Encounter | 153,515 |
| DocumentReference (includes clinical notes) | 153,515 |
| ExplanationOfBenefit | 153,515 |
| Procedure (includes MRI scans and EKGs) | 62,186 |
| Condition (includes relevant diagnoses) | 16,585 |
| Immunization | 12,281 |
| CareTeam | 6381 |
| CarePlan | 6381 |
| Medication | 1227 |
| MedicationAdministration | 1227 |
| ImagingStudy (includes DICOM images) | 3764 |
| Bundle | 1346 |
| Patient | 1346 |
| Provenance | 1346 |
| Device | 489 |
| AllergyIntolerance | 145 |
| Media (electrocardiogram images) | 1156 |

Health conditions rarely occur in isolation, and although the Coherent Data Set focuses on CVD, other health concerns that are associated with CVD are present in the synthetic patients. Both active and abated diagnoses are captured in Figure 7. The graph shows higher counts of active conditions that are commonly associated with CVD and that may

require a combination of imaging, genomics modeling, and physiologic values to diagnose. Additional graphs illustrating the demographic composition of synthetic patients in the Coherent Data Set can be found in Appendix A, Figures A1–A4. The figures show the distribution of patients by age group, race, sex, and whether they are deceased or living. The Coherent Data Set provides a more comprehensive view of the synthetic patients in that it can generate and combine all these forms of health information that would be present in a real patient's health record.

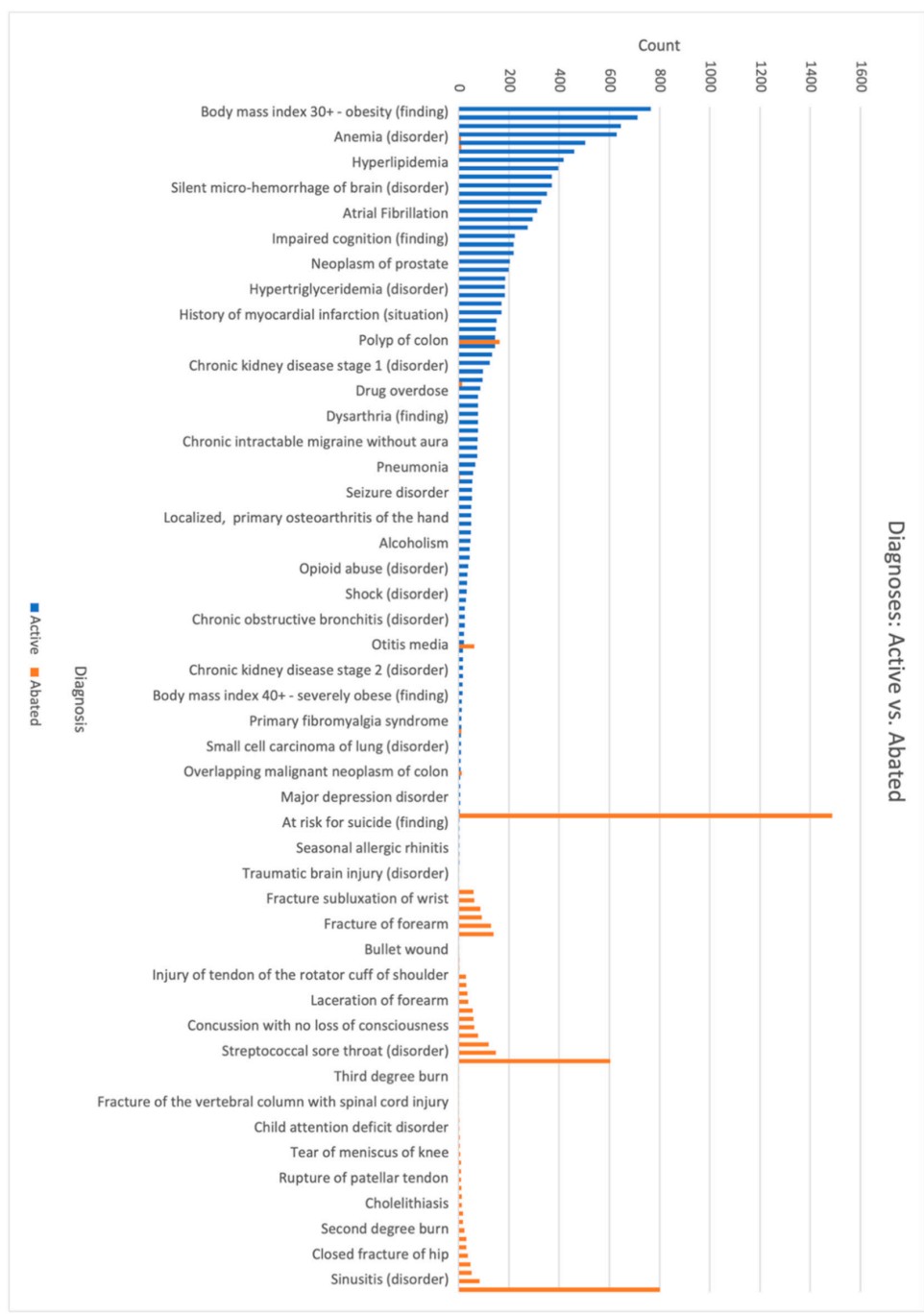

**Figure 7.** Distribution of active and inactive diagnoses among patients in the Coherent Data Set.

## 4. Discussion

### 4.1. Significance of the Coherent Data Set

We believe this paper and the associated synthetic data set is the first research that combines a multiplicity of synthetic data forms together in a single package and makes it publicly available as a digital public good. Synthetic genomic data derived from the Human Genome Project give each synthetic patient genetic risk markers related to cardiovascular disease. In the structured EHR data, the synthetic patients have corresponding clinical diagnoses for diseases they are genetically at risk for. Synthetic MRIs built from numerical phantoms illustrate cerebral microbleeds for those synthetic patients who suffered from strokes. Synthetic cardiovascular physiology data generated from hemodynamic models also demonstrate cardiovascular issues related to the genetic markers. Finally, simplified free text notes tie the data together.

Each data set was created using different methods suitable for each sub-domain based on literature relevant to each field of study (as described in the Materials and Methods section). The method to tie them together coherently is based on health information technology practices, using the HL7 FHIR international standard.

### 4.2. Validating the Results

Validating the synthetic data in the Coherent Data Set for clinical accuracy continues to be a challenge. A rigorous review process to test the accuracy of the synthetic data would require a singular real-world integrated data set of similar composition to compare the values. In the absence of another data set to compare with the complete records within the Coherent Data Set, individual components have been validated in their respective domains (reusing validated techniques and results from other researchers as cited and described in the Materials and Methods section), and the overall verification process focused on the structure of the data set. Continued collaboration between subject matter experts is critical for building the necessary knowledgebase to develop a rigorous validation process for future iterations of the Coherent Data Set.

Our team was unable to statistically validate the resulting synthetic data set against a real-world equivalent because a real-world equivalent is unavailable or does not yet exist. Validation against multiple smaller data sets was also not conducted due to reliance upon methods that have been previously validated or were based on established methodologies.

Therefore, the clinical accuracy of the results should be viewed with suspicion. However, the value proposition of the Coherent Data Set is in its ability to demonstrate relationships between data sources rather than clinical accuracy. Improving healthcare interoperability requires publicly accessible and readily available data with a realistic structure and accurate representation of concepts, not necessarily a high degree of statistical accuracy. We believe this resulting data set is suitable for the healthcare industry for the purposes of health interoperability development, testing, and integration. It is also suitable for educators and data scientists in the fields of health informatics and clinical informatics.

The Coherent Data Set is not suitable for Machine Learning or Natural Language Processing models, except for situations where researchers are bootstrapping work and building a processing pipeline prior to gaining access to whatever real data sets they might be targeting. However, just because a data set is not demonstrably accurate to a certain degree or metric does not mean it is not useful.

### 4.3. Alternative Methods

As a comparison to other synthetic data generation methods, the Coherent Data Set's various modeling and simulation approaches offer advantages over generative adversarial networks (GANs) and deep learning methods. For example, Synthea only leverages publicly available statistics to emulate patient life cycles, rather than depending upon real EHRs or patient data as training data for machine learning methods such as GANs [8]. Additionally, the modularity and extensibility of Synthea allow researchers to add new diseases or to add, remove, or customize the patient pathways using the Module-Builder's

(https://synthetichealth.github.io/module-builder/ (accessed on 5 March 2021)) point-and-click utility. These models are also fully explainable and interpretable; a researcher can easily review visual disease models to understand what is occurring, whereas models produced using deep learning methods are typically inscrutable. Finally, modeling and simulation allow for alternative scenarios to be developed and simulated, including what-if scenarios or scenarios where real data otherwise do not exist; such scenarios are outside the scope of deep learning models that are inherently limited to and bound by their training data. There is a potential cost to our approach, however, because GANs and deep learning methods may be able to tune their models to a higher degree of statistical accuracy. However, statistical accuracy alone is seldom the determining factor for usefulness, and the benefit of the Coherent Data Set's approach, which is more transparent and customizable, lends itself to supporting work on healthcare interoperability.

*4.4. Future Work*

Future work on the Coherent Data Set could take various directions. In its current form, the data set is a useful tool for illustrating interoperability without relying on de-identified patient information. With refinements and additions to the Coherent Data Set, it could be used by a broader audience for both health-related research and policy modeling. For example, incorporating data on social determinants of health could provide a clearer understanding of the overall health status of synthetic patients. Adding new data types such as consumer-generated data from wearable devices or voice data could also widen the scope of the Coherent Data Set and offer new ways to demonstrate interoperability or to support health-related research.

Additional research specific to genomic data, imaging, clinical notes, and physiology models or an expansion to include other diseases are also possible targets for future work. The data set began with a focus on clinical events and patient variables related to CVD, but the learnings and framework of the data set could be applied to other chronic illnesses. However, the generation of the Coherent Data Set required considerable time and computing power. Any future expansions to the data should be carefully evaluated for their potential impact and trade-offs in development and run time.

The Coherent Data Set can be downloaded at http://hdx.mitre.org/downloads/coherent-08-10-2021.zip (accessed on 31 March 2022).

**5. Conclusions**

The Coherent Data Set provides a publicly available set of data types that are woven together in a clinically coherent manner; it is possibly the only example of its kind. Using synthetic but realistic patient data for modeling interoperability and individual patient-level health status creates a functional, sandbox-like platform allowing researchers to work on health information interoperability, develop machine learning models (used as training data to demonstrate the art of the possible, not for production implementations), and perform non-clinical secondary research where realistic-but-not-real data are sufficient. Patient-level information that simulates data captured for an individual with CVD facilitates research without the potential privacy risks associated with the use of actual patient data. The Coherent Data Set is best suited for modeling and pipeline development or educational purposes, and it provides a framework that can be leveraged for health information exchange, research, and demonstrations.

**Supplementary Materials:** The following supporting information can be downloaded at: https://www.mdpi.com/article/10.3390/electronics11081199/s1, Table S1: SNPs linked to CV disease states.

**Author Contributions:** J.W., D.H., and K.M.B. contributed to the overall drafting of the paper as well as the physiology work, along with R.T.S., M.H.F., A.K., M.M., and J.H., who contributed to the genomics work while J.D., M.E.D., and G.E. developed the imaging data. B.W., L.L., and K.W. developed the clinical notes; M.H. and A.G. integrated the various data types together. F.X.C., E.G., and S.R. provided

clinical expertise across all topics, and Bates contributed to the coordination and facilitation of the writing process. All authors have read and agreed to the published version of the manuscript.

**Funding:** The research reported in this publication was supported by the MITRE Innovation Program. Approved for public release; distribution unlimited. Case number 21-2657.

**Data Availability Statement:** The Coherent Data Set can be downloaded here: http://hdx.mitre.org/downloads/coherent-08-10-2021.zip (accessed on 31 March 2022).

**Conflicts of Interest:** The authors declare no conflict of interest.

## Abbreviations

| Abbreviation | Term |
| --- | --- |
| CT | Computed Tomography |
| CVD | Cardiovascular Disease |
| ECG | Electrocardiogram |
| EHR | Electronic Health Record |
| FASTA | Text-based format for representing either nucleotide sequences or amino acid sequences |
| FHIR | Fast Health Interoperability Resources |
| FLAIR | Fluid Attenuated Inversion Recovery |
| GMF | Generic Module Framework |
| LSTM | Long Short-term Memory |
| LVEF | Left Ventricular Ejection Fraction |
| MRI | Magnetic Resonance Imaging |
| SBML | Systems Biology Markup Language |
| SNP | Single Nucleotide Polymorphism |
| TIA | Transient Ischemic Attack |
| V-fib | Ventricular Fibrillation |

## Appendix A  Summary Statistics for The Coherent Data Set

**Table A1.** 172 synthetic patients have three different data file types, while 289 patients do not have a DICOM image or DNA file to accompany their FHIR Bundle.

| Data Type | Count |
| --- | --- |
| DICOM Image, DNA Data, and FHIR Bundle | 172 |
| No DICOM Image or DNA File | 289 |

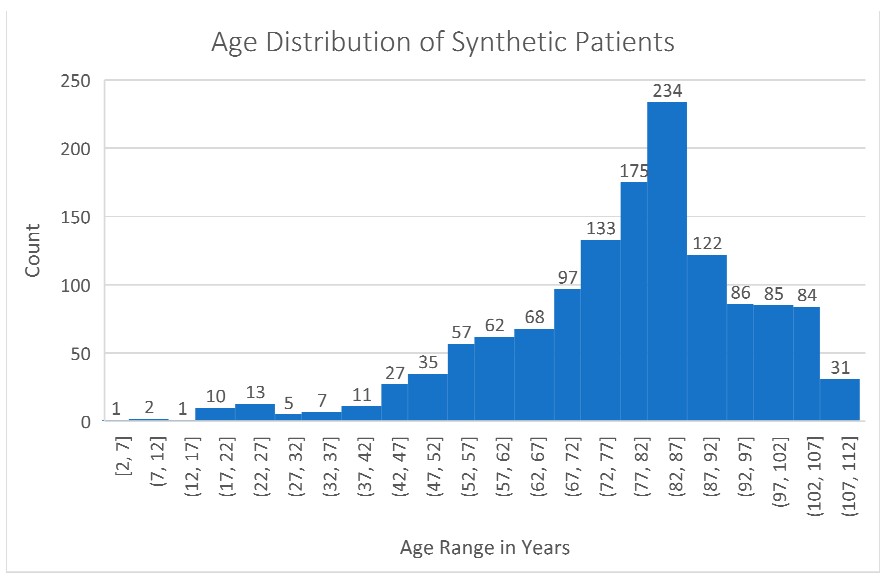

**Figure A1.** Distribution of synthetic patients by age grouping.

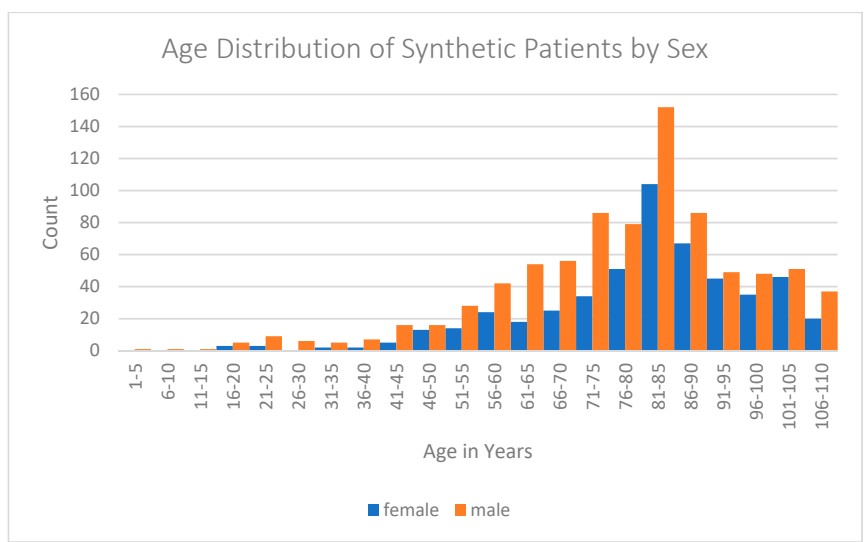

**Figure A2.** Distribution of synthetic patients by age group and sex.

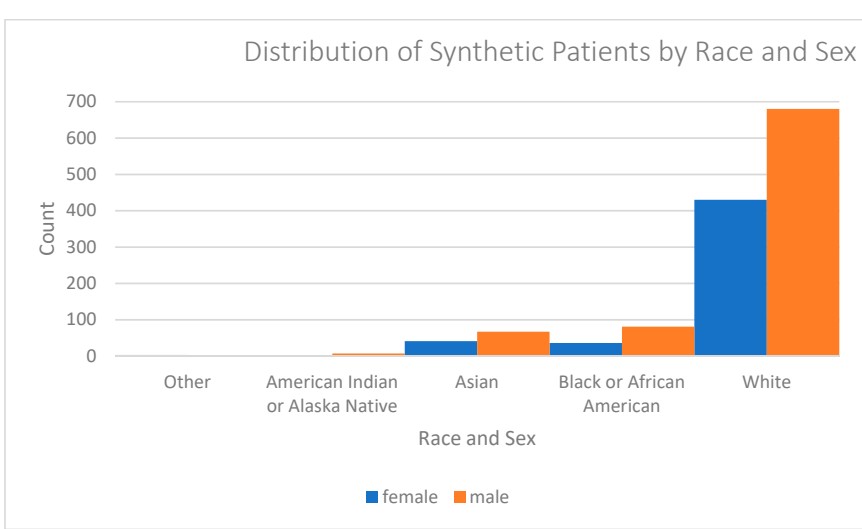

**Figure A3.** Distribution of patients by race and sex.

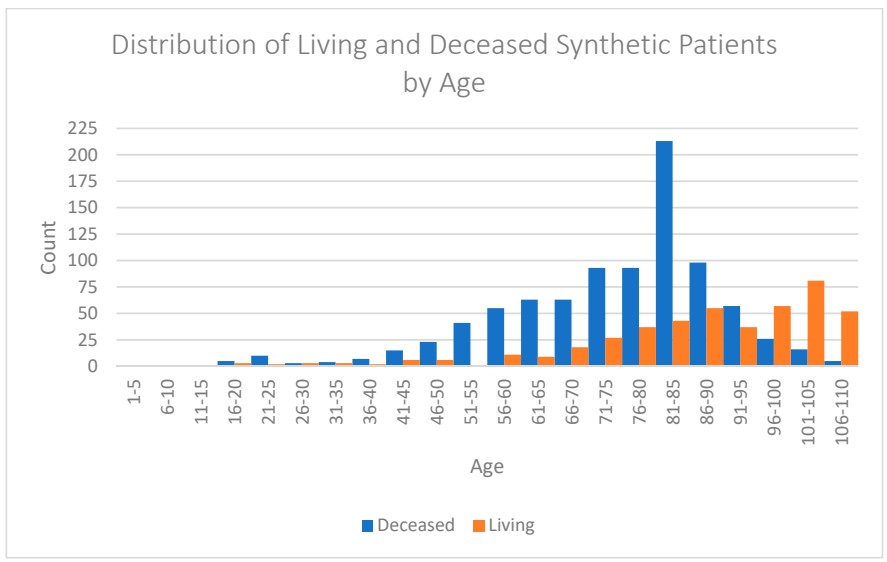

**Figure A4.** Distribution of deceased and living patients by age group.

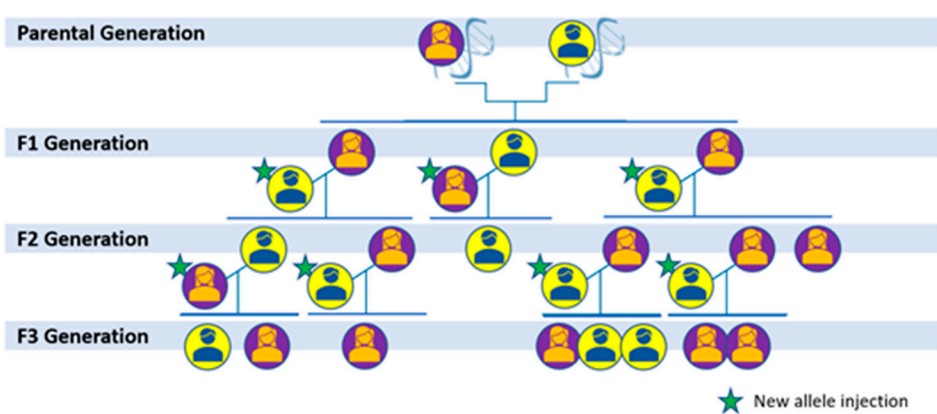

**Figure A5.** Generation of familial synthetic genomes.

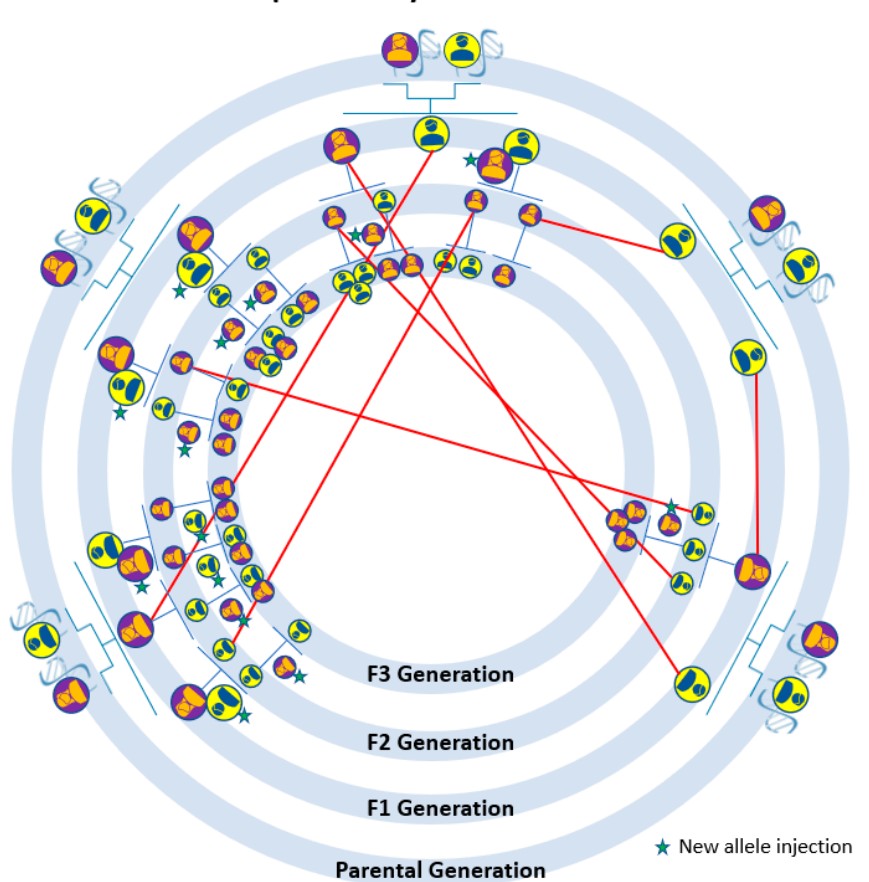

**Figure A6.** Generation of an entire population of synthetic genomes.

**(I.)**

**(II.)**

**(III.)**

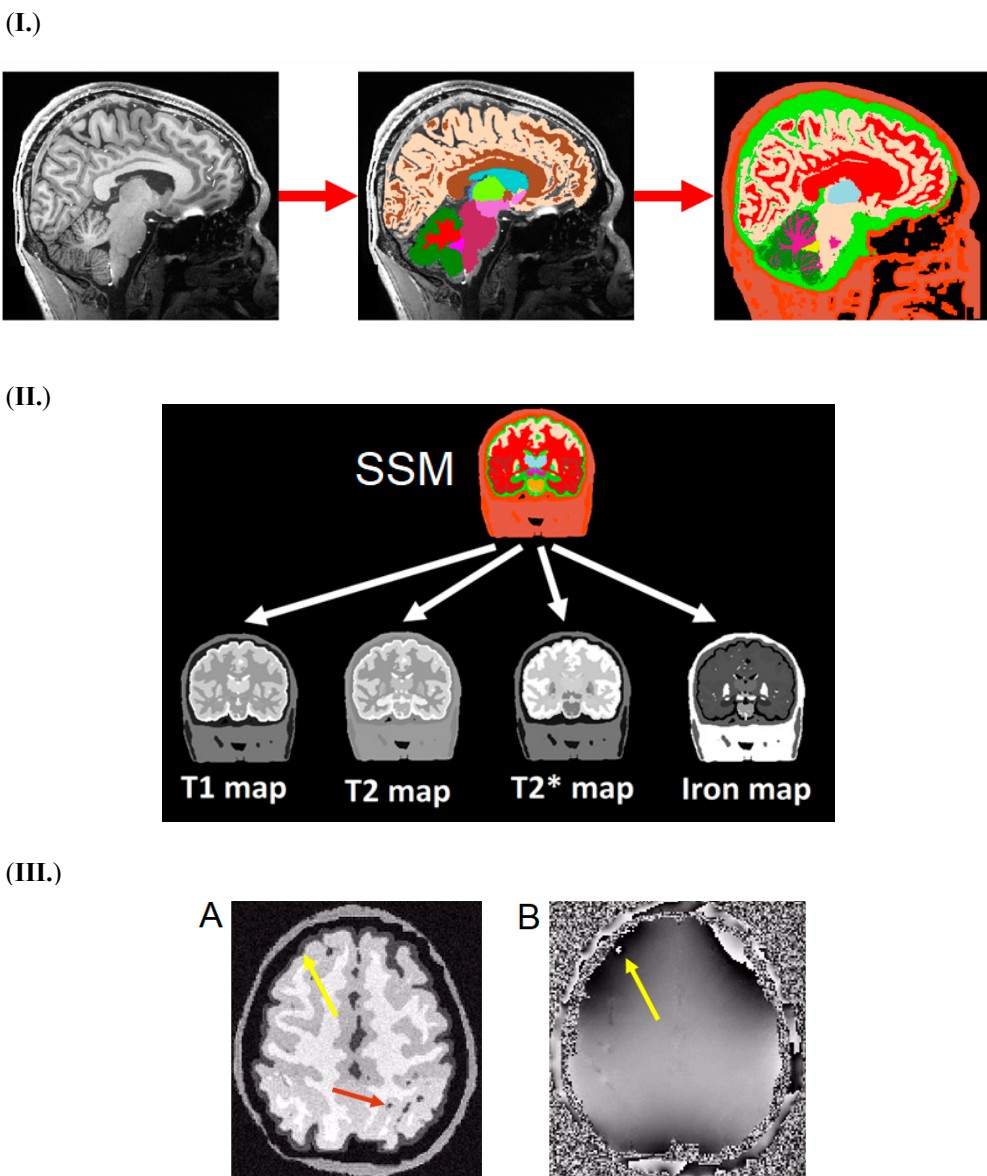

**Figure A7. (I)** Root Segmentation Map (RSM) process. Colors indicate unique segmented brain and head regions. **(II)** Applying unique EMTP tissue properties. Each segment has a defined range of ground truth EMTP tissue properties that are selected when creating the SSM. **(III)** MRI Simulator output. A shows the magnitude image, and B shows the phase image. The yellow arrow indicates a microbleed, while the red arrow indicates arterial vessels. The microbleed is easily detected in the phase, as opposed to the magnitude image.

1998-05-02

# Chief Complaint
- Hunger
- Blurred Vision
- Tingling in Hands and Feet
- Thirst
- Fatigue

# History of Present Illness
Wesley533
is a 68 year-old non-Hispanic white male. Patient has a history of fracture of forearm.

# Social History
Patient is married. Patient is an active smoker and is an alcoholic.

Patient identifies as heterosexual.

Patient comes from a middle socioeconomic background.

Patient did not finish high school.

Patient currently has Medicare.

# Allergies
No Known Allergies.

# Medications
amlodipine 5 mg oral tablet; insulin human, isophane 70 unt/ml / regular insulin, human 30
unt/ml injectable suspension [humulin]; simvastatin 20 mg oral tablet; ibuprofen 200 mg oral
tablet; clopidogrel 75 mg oral tablet; nitroglycerin 0.4 mg/actuat mucosal spray

# Assessment and Plan
Patient is presenting with stroke.

## Plan

The following procedures were conducted:
- echocardiography (procedure)
- percutaneous mechanical thrombectomy of portal vein using fluoroscopic guidance
The patient was prescribed the following medications:
- alteplase 100 mg injection

**Figure A8.** Clinical note from the encounter for a synthetic patient experiencing a stroke, which triggered genetic testing.

2001-12-21

# Chief Complaint
- Decreased in Judgement
- Memory Loss
- Decreased Visual Perception

# History of Present Illness
Abbey813

is a 66 year-old non-hispanic white female.

# Social History
Patient is married. Patient is an active smoker and is an alcoholic.

Patient identifies as heterosexual.

Patient comes from a middle socioeconomic background.

Patient has a high school education.

Patient currently has Medicare.

# Allergies
No Known Allergies.

# Medications

**Figure A9.** Clinical note for a synthetic patient that received an MRI, which found a micro-hemorrhage of the brain.

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
