# Peer review of "The “Coherent Data Set”: Combining Patient Data and Imaging in a Comprehensive, Synthetic Health Record"

_electronics, doi:10.3390/electronics11081199_

Round 1
Reviewer 1 Report
The paper introduces a comprehensive synthetic database that combines clinical, imaging, genomic and physiological information. Although the topic is of interest and importance, it suffers flaws in its introduction, and the data generating methods are not based on established work. No validation is offered to assess how realistic the dataset is. Major comments:- The manuscript is not written by acceptable standards of scientific writing: eg, introduce the topic by providing background on the problem, proposed solutions, the proposed new solutions, the purpose of the paper and how it is organized.
- The paper addresses a highly challenging task of generating a comprehensive dataset, by methods that do not seem to be based on scientifically sound algorithms
- . In particular, generating synthetic genomic data can be a study by itself - is there a reference ? were the methods used validated?
- Has the realistic aspect of the database validated? In particular, how were correlations between various variables, and more challengingly, associations between genomic data and disease, addressed within the data generating process? For instance, one way to validate the data and make sure it imitates real data is to compare results from analysis based on real data to results based on the proposed synthetic data (see Reiner-Benaim et al, 2020: Analyzing Medical Research Results Based on Synthetic Data and Their Relation to Real Data Results: Systematic Comparison From Five Observational Studies
- If the proposed data is focused on cvd - it should be mentioned in the manuscript title and abstract. Also, why was cvd chosen, and how it projects on other diseases that will be joined in the future?
Author Response
The paper introduces a comprehensive synthetic database that combines clinical, imaging, genomic and physiological information. Although the topic is of interest and importance, it suffers flaws in its introduction, and the data generating methods are not based on established work. No validation is offered to assess how realistic the dataset is. Major comments:
- The manuscript is not written by acceptable standards of scientific writing: eg, introduce the topic by providing background on the problem, proposed solutions, the proposed new solutions, the purpose of the paper and how it is organized.
Author Response:
We acknowledge that our introduction lacked sufficient background on the problem, and related prior work, which we have corrected in the Introduction. We hope these changes address the concern.
- The paper addresses a highly challenging task of generating a comprehensive dataset, by methods that do not seem to be based on scientifically sound algorithms
Author Response:
We respectfully disagree with this Reviewer’s generalized statement.
- In particular, generating synthetic genomic data can be a study by itself - is there a reference ? were the methods used validated?
Author Response:
Methods used to select and attribute SNPs linked to cardiovascular disease within the algorithm were chosen based on their validation described within the scientific literature. For directing the reader to these SNP descriptions and their respective references, we have included a supplementary table characterizing each SNP used within the study, providing their reference SNP cluster IDs.
- Has the realistic aspect of the database validated? In particular, how were correlations between various variables, and more challengingly, associations between genomic data and disease, addressed within the data generating process? For instance, one way to validate the data and make sure it imitates real data is to compare results from analysis based on real data to results based on the proposed synthetic data (see Reiner-Benaim et al, 2020: Analyzing Medical Research Results Based on Synthetic Data and Their Relation to Real Data Results: Systematic Comparison From Five Observational Studies
Author Response:
Methods used to select and attribute SNPs linked to cardiovascular disease within the algorithm were chosen based on their validation described within the scientific literature. For directing the reader to these SNP descriptions and their respective references, we have included a supplementary table characterizing each SNP used within the study.
The physiology portion of the work reused peer-reviewed and verified models (McSharry et al., Smith et al) and was validated against Hosseini et al. then inspected by clinician SMEs for the corresponding diseases each synthetic patient presented with, corresponding to the linked genomic data. We have no access to a real-world data set that links genomic data with ECG phenotypes to show mathematical significance, so we must rely on expertise.
The underlying Synthea data itself has been validated and/or repudiated repeatedly in the literature so there is little point in re-examining that. However, we will point out that unlike nearly every other synthetic health data generation method or product or publication, Synthea provides access to all the source code, inputs, models, and synthetic data so that the scientific community may reproduce our work and has the tools to validate or repudiate it. You cannot say the same thing about any other serious solution that has been peer-reviewed.
- If the proposed data is focused on cvd - it should be mentioned in the manuscript title and abstract. Also, why was cvd chosen, and how it projects on other diseases that will be joined in the future?
Author Response:
We appreciate this recommendation from the reviewer to include CVD in the title and abstract. CVD was selected because it is a commonly occurring disease with a high prevalence rate, with severe outcomes, and represents a significant global burden. However, since CVD was used as an example to convey the many different data types of the Coherent Data Set and not to advance the research of CVD, we have chosen not to modify the title. Similarly, we are limited by the maximum word count for the abstract and we want to ensure the focus remains on the methods and purpose of the data set in the brief passage.
Reviewer 2 Report
-This paper introduces the publicly available "Coherent Data Set", which combines patient data and imaging in a synthetic health record. It is focused on cardiovascular disease (CVD).
-The areas considered in the "Coherent Data Set" are: genomics, imaging, clinical notes, and physiology.
-The authors should have included a background or literature review section mentioning similar datasets to the "Coherent Data Set", or to individual data sets focused on the considered areas (genomics, imaging, clinical notes, and physiology).
-While the different areas considered for the "Coherent Data Set" are well described, it is not clear to me how the authors validated the "Coherent Data Set". It is mentioned that individual components were validated but there are no results presented in the form of tables or charts.
-From the synthetic patient data shown in Figures A7, A8, A9 and A10, did you consider the balance of the data on how it is distributed in a real data set? Would a synthetic data set benefit from a more balanced data set for the purpose of data analysis? The distribution of patients by race and sex is particularly unbalanced, does that reflects real data sets that you studied and based your data set generation approach?
Author Response
-This paper introduces the publicly available "Coherent Data Set", which combines patient data and imaging in a synthetic health record. It is focused on cardiovascular disease (CVD).
-The areas considered in the "Coherent Data Set" are: genomics, imaging, clinical notes, and physiology.
-The authors should have included a background or literature review section mentioning similar datasets to the "Coherent Data Set", or to individual data sets focused on the considered areas (genomics, imaging, clinical notes, and physiology).
Author Response:
We added some background information on previous work that has been done with synthetic health data to the introduction portion of the paper, including research focusing on specific types of data (e.g., genomics, notes, imagery, physiology) as well as previous attempts to link together heterogeneous data types.
-While the different areas considered for the "Coherent Data Set" are well described, it is not clear to me how the authors validated the "Coherent Data Set". It is mentioned that individual components were validated but there are no results presented in the form of tables or charts. -From the synthetic patient data shown in Figures A7, A8, A9 and A10, did you consider the balance of the data on how it is distributed in a real data set? Would a synthetic data set benefit from a more balanced data set for the purpose of data analysis? The distribution of patients by race and sex is particularly unbalanced, does that reflects real data sets that you studied and based your data set generation approach?
Author Response:
Regarding the (un)balance of data: this is a great point. Equity and representation in data, and health data in particular is an important topic. Research has shown gender and racial disparities in CVD.
For one example, see Kanchi R et al. Gender and Race Disparities in Cardiovascular Disease Risk Factors among New York City Adults: New York City Health and Nutrition Examination Survey https://doi.org/10.1007/s11524-018-0287-x
This real-world disparity is also true in our synthetic data, and this is exacerbated by the fact that we used Massachusetts demographics, which has a small minority population.
Regarding validation, please see our response to item #4 from Reviewer #1.
Reviewer 3 Report
In this manuscript, the authors discussed the the Coherent Data Set, a publicly available set of data types that are woven together in a clinically coherent manner, to demonstrate that the Coherent Data Set is best suited for modeling and pipeline development, or educational purposes, providing a framework that can be leveraged for health information exchange, research, and demonstrations.
The manuscript was written with a lot of details and explanations. I have a couple of comments before it can be considered for publication:
- In the introduction part, could the authors explain more clearly what improvements they have made on this data compared with previous studies? For example, did they clean up the data to make it more organized? Or did they do some quantitative analysis to demonstrate that this dataset can be utilized for some new experiments which have not been done before?
- Could the authors provide some figures to illustrate the feature of the data? For example, histograms or distributions, sample sizes, feature space, etc.
Author Response
In this manuscript, the authors discussed the the Coherent Data Set, a publicly available set of data types that are woven together in a clinically coherent manner, to demonstrate that the Coherent Data Set is best suited for modeling and pipeline development, or educational purposes, providing a framework that can be leveraged for health information exchange, research, and demonstrations.
The manuscript was written with a lot of details and explanations. I have a couple of comments before it can be considered for publication:
- In the introduction part, could the authors explain more clearly what improvements they have made on this data compared with previous studies? For example, did they clean up the data to make it more organized? Or did they do some quantitative analysis to demonstrate that this dataset can be utilized for some new experiments which have not been done before?
Author Response:
We edited our introduction to emphasize that we didn’t necessarily clean the data or perform a quantitative analysis, but instead we combined several types of synthetic data in a new way and are making it publicly available for research and work related to Health Interoperability (or the lack thereof). This unique combination forms a synthetic data set with many types and formats of patient data that is the first of its kind, that we know of.
- Could the authors provide some figures to illustrate the feature of the data? For example, histograms or distributions, sample sizes, feature space, etc.
Author Response: We moved some of the summary statistics-related tables and a graph of active and inactive diagnoses among patients from the appendix to the Results section of the paper.
Reviewer 4 Report
The topic is very interesting.
The results and discussion sections should be improved.
Explain the key findings in a more comprehensive manner.
Some sentences need to be rewritten.
Author Response
The topic is very interesting.
The results and discussion sections should be improved.
Author Response: We attempted to improve these sections.
Explain the key findings in a more comprehensive manner.
Author Response: We revised the introduction and the conclusions in attempt to do so.
Some sentences need to be rewritten.
Author Response: We agree, and we rewrote several sentences during the revision process.
Round 2
Reviewer 1 Report
Some of the previous comments have not been addressed, and some new comments are added:
- The Introduction Section is meant to provide background, reference, problem, and state the purpose and organization of the paper. please end the Introduction by clearly describing the
(1) purpose of the paper - is it describing the Coherent Data Set? the data it includes? the algorithms that generate it? available packages? etc.
(2) what is done in the paper to achieve this purpose?
(3) how is the paper organized? - particularly, what is presented in the Methods? the results in the Results Section - what are they results of? - The detailed description of the coherent data set should be in the Methods Section. Please create a separate sub-section titled "The Coherent Data Set" (within the Methods), where the dataset is described in details. Within in the Introduction, it should only be generally and breifly described.
- Regarding added sentence: "In the absence of another data set to compare the complete records within the Coherent Data Set, individual components have been validated in their respective domains..." - please provide references.
- This whole newly added paragraph ("Validating the synthetic data in the Coherent Data Set for clinical accuracy continues to be a challenge") should be moved to the Discussion.
- Within the Discussion: Please state clearly the benefit from the Coherent Data Set, and for whom (medical/biology researchers? data scientists? health care managers and policy makers?) Particularly in light of this added sentence: "Improving healthcare interoperability requires publicly accessible and readily available data with realistic structure and accurate representation of concepts, not necessarily a high degree of statistical accuracy."
- In author response: "CVD was selected because it is a commonly occurring disease with a high prevalence rate, with severe outcomes, and represents a significant global burden. " - please specify that in the Introduction, within the means to achieve the purpose of the paper (see comment 1(2) above).
Author Response
1. The Introduction Section is meant to provide background, reference, problem, and state the purpose and organization of the paper. please end the Introduction by clearly describing the 
(1) purpose of the paper - is it describing the Coherent Data Set? the data it includes? the algorithms that generate it? available packages?  etc.
(2) what is done in the paper to achieve this purpose?
(3) how is the paper organized? - particularly, what is presented in the Methods? the results in the Results Section - what are they results of?
Author Response:
We added a paragraph to the end of the introduction to address the purpose, what was achieved, and the paper organization.
2. The detailed description of the coherent data set should be in the Methods Section. Please create a separate sub-section titled "The Coherent Data Set" (within the Methods), where the dataset is described in details. Within in the Introduction, it should only be generally and breifly described. 
Author Response:
We moved details of the data set from the introduction into the Methods section and reorganized it for clarity.
3. Regarding added sentence: "In the absence of another data set to compare the complete records within the Coherent Data Set, individual components have been validated in their respective domains..." - please provide references. 
Author Response:
We cited these references in the individual Materials and Methods subsections, so we added a note to highlight this.
4. This whole newly added paragraph ("Validating the synthetic data in the Coherent Data Set for clinical accuracy continues to be a challenge") should be moved to the Discussion.
Author Response:
We moved the paragraph to the Discussion.
5. Within the Discussion: Please state clearly the benefit from the Coherent Data Set, and for whom (medical/biology researchers? data scientists? health care managers and policy makers?) Particularly in light of this added sentence: "Improving healthcare interoperability requires publicly accessible and readily available data with realistic structure and accurate representation of concepts, not necessarily a high degree of statistical accuracy." 
Author Response:
We added a paragraph in the discussion, and generally reorganized the section to consolidate bits of discussion that were previously spread about.
6. In author response: "CVD was selected because it is a commonly occurring disease with a high prevalence rate, with severe outcomes, and represents a significant global burden. " - please specify that in the Introduction, within the means to achieve the purpose of the paper (see comment 1(2) above).
Author Response:
Acknowledged and added just prior to the final purpose paragraphs.
Reviewer 2 Report
I think the authors addressed all my comments appropriately and in the best way they could. This has resulted in an improved version of the paper, which I think should be accepted.
Author Response
I think the authors addressed all my comments appropriately and in the best way they could. This has resulted in an improved version of the paper, which I think should be accepted.
Author Response: Thank you for your advice and thorough review of the paper!